# Hurricane disturbance results in positive effects on tropical stream meiofauna abundance and richness

Josué Santiago-Vera[1]☯, Alonso Ramírez[ID][2]☯*

1 Department of Natural and Applied Sciences, Mount Mercy University, Cedar Rapids, Iowa, United States of America, 2 Department of Applied Ecology, North Carolina State University, Raleigh, North Carolina, United States of America

☯ These authors contributed equally to this work.
* alonso.ramírez@ncsu.edu

## Abstract

Hurricanes are major natural disturbances that significantly influence tropical ecosystems. While most research focuses on large-bodied organisms, understanding the impact of hurricanes on small-bodied biota, such as meiofauna, is crucial, especially as climate change models predict more frequent and intense storms. Puerto Rico offers a unique setting to study these effects, as hurricanes and tropical storms are frequent. This research examined the short- (Post-Hurricane A) and medium-term (Post-Hurricane B) impacts of Hurricanes Irma and María (September 2017) on stream meiofaunal communities in a tropical stream in the Luquillo Experimental Forest. Twelve samples were collected monthly from pools across two stream reaches for eight months before and after the hurricanes. Environmental variables, such as discharge, sediment composition, and biotic data, were recorded. Meiofauna were identified to the lowest possible taxonomic level and categorized by phyla and feeding groups. Results showed a significant increase in both richness and abundance of meiofauna following the hurricanes. Richness and abundance peaked during Post-Hurricane A and declined slightly during Post-Hurricane B. This trend was linked to an increase in coarse particulate organic matter, potentially the result of defoliation and debris from hurricane damage of riparian forest. Community structure analyses revealed significant differences between pre- and post-hurricane periods. Variables such as stream discharge, macroinvertebrate presence, sediment size, and shrimp abundance were all influenced by hurricane impacts and correlated with changes in meiofaunal communities. During Post-Hurricane A, models explaining meiofaunal variation involved variables associated with ecosystem disturbance. In contrast, Post-Hurricane B models were simpler, suggesting a level of ecological stabilization. These findings suggest that hurricanes influence meiofaunal communities, but that these organisms are likely benefiting from hurricane disturbance. Given expected increases in hurricane activity due to climate change, hurricanes may play

provided the original author and source are credited.

**Data availability statement:** Meiofauna community data before the hurricane is available at https://doi.org/10.5281/zenodo.3879232. Meiofauna community data after the hurricane is available at https://doi.org/10.5281/zenodo.18393393. Environmental data can be found at the Luquillo LTER database, at https://luquillo.lter.network/long-term-datasets/ and under the section entitled "Stream Flow Reduction Experiment (StreamFRE)".

**Funding:** Support for this work was provided by the National Science Foundation via grant DEB-1831952 (Luquillo Long-Term Ecological Research Program). There was no additional external funding received for this study.

**Competing interests:** The authors have declared that no competing interests exist.

a long-term role in shaping the structure and function of tropical stream communities, in particular for small-body size organisms.

## Introduction

Hurricanes (cyclones and typhoons in the Indian and South Pacific oceans and the Northwest Pacific Ocean) are natural and important disturbances that shape ecosystem structure and dynamics in many regions (e.g., the Caribbean, Asia, and Australia) [1–7]. Ecosystems regularly affected by hurricanes are composed of species that are resilient (i.e., recover quickly from disturbance) or resistant (i.e., change little with disturbance) [4,6,8–10]. For example, while tree species with dense wood trunks are resistant to hurricanes [11,12]. mangrove species (e.g., *Avicennia germinans* and *Laguncularia racemosa*) tend to be resilient, recovering quickly from structural damage via epicormic sprouts [13]. Our understanding of hurricane impacts on vegetation is, in general, more complete than other groups. We know less about faunal responses, in particular small aquatic organisms that inhabit the sediments (i.e., meiofauna).

Meiofauna are small-bodied heterotrophic organisms often associated with benthic sediments in aquatic ecosystems [14–17]. These organisms form an important energy link between fine basal resources, like fine organic matter, and aquatic macroinvertebrates. Meiofauna is strongly affected by streamflow, substratum composition, and the availability of organic matter [18,19]. For instance, Gaudes and colleagues [20] found that meiofauna in the Mediterranean are adapted to seasonal stream floods. Thus, a specific community of meiofauna is present in each season. Moreover, the substratum type (based on sediment size) is a predictor of meiofaunal community composition and abundance in temperate streams [18,21]. High organic matter was also related to high meiofaunal abundance [21]. As hurricanes are known to impact some of those variables (e.g., discharge [22–24], substratum [25,26], and organic matter [27]), major effects on stream meiofauna could be expected.

Puerto Rico offers a unique opportunity to understand hurricane impacts as the island is regularly impacted by these atmospheric events [1,10,28]. The recurrence time of hurricanes for Puerto Rico is between 22 and 62 years [29]. However, due to the occurrence of recent hurricanes, this recurrence period has shortened to approximately 42 years [30]. In addition to major hurricanes, low-intensity hurricanes have a four-year frequency [31] and tropical storms occur every year. The Luquillo Long-Term Ecological Research Program within the Luquillo Experimental Forest has gathered data on hurricane impacts since 1989 [10]. Much of our understanding of hurricane effects on Puerto Rican ecosystems comes from studies of terrestrial communities (e.g., vegetation, snails, birds) [5,10]. There is less information on aquatic communities [32,33], but none on stream meiofauna.

We aim to assess the short- and medium-term effects of hurricane impacts on meiofaunal communities in a tropical stream in Puerto Rico. Hurricane disturbances have strong impacts on stream ecosystems, with consequences that last for several months to several years [10,34,35]. Immediate effects are associated with the heavy

rains that result in major floods, which are a source of mortality to aquatic fauna and a pulse of organic matter due to forest canopy defoliation [32,33]. Canopy removal increases solar radiation in the channel, increasing algal growth, and generates effects that are evident a few months after the impact (medium-term effects). Our objective is to describe those changes and relate them to environmental factors altered by hurricane disturbance. We expected meiofauna richness and abundance to decrease in response to hurricane disturbance. We also expect positive responses by meiofaunal groups that could benefit from post-hurricane conditions (e.g., increased fine particles, higher algal biomass). Changes in the sediment characteristics (e.g., proportion of fine particles) are predicted to be major drivers of changes in meiofauna.

## Methods

### Study site

The study site was Quebrada Prieta (18°19´20´´N, 65°48´50´´W), a tributary of the Espíritu Santo watershed in the Luquillo Experimental Forest (LUQ), also known as El Yunque National Forest, Puerto Rico. Quebrada Prieta has two main tributaries: Quebrada Prieta A and Quebrada Prieta B (QPA and QPB, respectively, hereafter). We selected a 100 m section at each tributary for the study, at ~400m in elevation, and chose nine pools within each reach for the study. Before hurricane disturbance, both reaches were heavily covered by an evergreen canopy dominated by *Dacryodes excelsa* and *Prestoea montana* [36]. The geomorphology is typical of other streams in the zone, characterized by low sinuosity and a steep-gradient, with the channel boulder dominated and some areas with sand [37,38]. The climate is characterized by a relatively stable mean monthly air temperature, ranging from 21 to 26°C with an annual mean of 23°C, and a mean monthly rainfall between 200–400 mm [39,40]. This forest is aseasonal, though the months between September and December could be wetter [10,41].

### Hurricanes Irma and María

Hurricanes Irma and María impacted LUQ in September 2017. Hurricane Irma developed from a tropical wave that by August 29 was a tropical storm. On September 1, Irma was an intense hurricane, and by September 5, it was a powerful category five hurricane (Saffir-Simpson Hurricane Wind Scale). The eye of Irma did not make landfall in Puerto Rico, passing 80 km north of the island between September 6 and 7. Tropical storm winds and heavy rainfall affected the northeastern part of Puerto Rico, and total rainfall estimates between 254 and 381 mm were reported [41].

Hurricane María impacted Puerto Rico two weeks after Irma. It developed from a tropical wave. On September 16, it was a tropical depression, and later that same day, it became a tropical storm. On September 17, María was a hurricane, and by September 18, it was a category five hurricane. On September 20, it made landfall as a powerful category four hurricane, entering Yabucoa in the southeast of Puerto Rico. Hurricane winds were detected on the entire island. After several hours, on September 20, the hurricane eye exited through the northwestern side of Puerto Rico. Rainfall was extreme, with stations recording up to 965 mm [42]. In less than one month, two extreme events impacted Puerto Rico, and an opportunity to understand hurricane effects on tropical stream meiofauna arose.

### Sampling of meiofauna

Meiofauna is defined by body size, though the size delimitation varies among authors [14,16,43,44]. Here, we focused on heterotrophic organisms living in aquatic sediments between 500 µm and 42 µm, with 42 µm being the smallest limit proposed [43]. This definition includes protists and the early instars of aquatic insects [14,43]. Each month, six of the nine pools were randomly selected for sampling in each tributary for a total of 12 pools per month. Due to conditions following the September 2017 hurricanes, sampling was missed in September 2017. However, the first post-hurricane sampling occurred within 30 days after Hurricane María impacted the zone. This resulted in eight months of pre-impact data and eight months of post-impact data.

Standard meiofaunal sampling requires the use of corers to sample sediment accumulations [15,45], but the rocky substrate of Prieta made this impossible. We designed a sampling methodology adapted to our study site [46]. Available habitats in Prieta were small and shallow pools with sediments, and we sampled them by pipetting the sediments instead of coring, using a 0.0062 m$^2$ polyvinyl chloride (PVC) ring to delimit the area and using a pipet with an aperture of 5 mm. The sampler was randomly inserted (1.5 cm depth) in pool areas with sediment. Each sample consisted of 290 mL of sediment and water collected within the PVC ring and 10 mL of 50% cold glutaraldehyde, resulting in 300 mL of sample fixed in 2% glutaraldehyde [47] and stored until processing.

Species richness and abundance were calculated from a 10 mL (3% of the sample) subsample after homogenizing the sample [46]. Meiofauna was separated via density gradient [48] and filtered through a 25 mm diameter nitrocellulose gridded filter (1.2 µm pore size). The density gradient method consists of using Ludox®. This colloidal silica solution generates a density gradient when centrifuged at 4,300 g and allows for the extraction of organisms. The method is expected to extract up to 99% of the microorganisms present in sediments [48]. Since samples were from freshwater, the desalination steps in were omitted. Samples from the pre-hurricanes period were stained with a quantitative Protargol staining (QPS) [47–49]. The QPS staining followed the modification suggested by colleagues [49] to stain the samples. The QPS requires gold chloride, which has to be stored in temperatures between 2–8° C. Post-hurricane samples were treated differently due to the lack of electric power in Puerto Rico for several months. We stained samples with Rose Bengal (0.1 g Rose Bengal/200 mL 5% formalin) [50] as this can be stored at room temperature. The two staining methods allow us to observe and identify meiofaunal organisms. Meiofauna individuals were identified to the lowest taxonomic level possible, otherwise we group them into morphospecies. Filters were thoroughly screened, and all organisms were measured (AmscopeTM Microscope Camera SKU: MU1000). For richness and abundance estimates, we counted organisms that measured between 42 and 500 µm. Identifications of groups like nematodes, ciliates, testate amoebas were conducted under a total magnification of 1000X. In the case of nematodes, morphospecies were assigned based on oral structures. All meiofauna were classified into major trophic guilds based on literature [51–55] and evidence of organisms inside their guts when observing the slides.

## Environmental variables

Sediment analysis was performed with the remaining volume of the samples. This volume was sieved to separate particles by size classes: coarse sand, medium sand, and fine sand. We chose these sizes because the dominant small particles reported for our study site are mostly sands [37]. After sieving, samples were dried at 70˚ C for two days, re-weighted, and ashed at 500˚C for 1 hour to calculate AFDM. Percentages of sediment by size class (coarse, medium, and fine sand) were calculated and used in statistical analyses.

Stream discharge (L/s) was estimated using a known relationship between water level and stream discharge for each reach. Water level was measured continuously in Quebrada Prieta using a pressure transducer (Onset Computers, HOBO U20L-04) [56]. This data logger measures water level using barometric pressure every 15 minutes. Loggers were placed in the lowermost sampling pools of QPA and QPB. Daily data were averaged 30 d before the meiofauna sampling date for our statistical analysis.

The study site is part of a long-term research location (LUQ-LTER); thus, additional data were available to relate to meiofaunal communities. Available data include water chemistry, shrimp density, macroinvertebrate assemblages, chlorophyll, and coarse organic matter input. Water chemistry was collected weekly and analyzed by the Water Quality Analysis Laboratory at the University of New Hampshire. Colorimetric methods were used to determine nitrogen and silica concentrations. Shrimps, macroinvertebrates, and algal biomass were collected monthly in the same study pools. Shrimps were sampled with baited traps left in the pool overnight. Macroinvertebrates were collected using corer samplers (0.0314 m$^2$). The collected material was fixed with formaldehyde (37%). Separation of organisms from sediment and organic matter was done in the laboratory. Collected macroinvertebrates were placed in vials with 80% ethanol and

later identified to the family level. Chlorophyll was estimated monthly as mean chlorophyll concentrations from Loeb samplers. In each pool, 7 Loeb samples were collected from rock surfaces. Detailed methods for the LUQ-LTER sampling are found in [39]. The meiofaunal samples were collected one week after the macroinvertebrate sampling in the same study pools.

## Statistical analyses

We assessed patterns of richness, abundance, and community structure of meiofauna before (January to August 2017) and after the hurricanes (October 2017 to May 2018). The post-hurricane period was divided into two subperiods: immediately after the hurricane (October to December 2017, hereafter Post-Hurricane A), and medium-term after the hurricane (January to May 2018, hereafter Post-Hurricane B). These three periods were delimited based on rainfall patterns and time from disturbance. The post-hurricane A period corresponded to the remainder of the hurricane season. Post-hurricane B corresponded to the dry season.

We compared richness and abundance among periods using a Kruskal Wallis test due to lack of normality according to a Shapiro-Wilks test (richness: $W = 0.87$ $p = 0.001$; abundance: $W = 0.78$, $p = 0.001$). Pairwise comparisons of means were made if significant differences were found. We removed from the analysis pools with zero individuals, as slides with no individuals were assumed to be the result of methodological difficulties during the Protargol staining procedure. Some samples were rich in organic matter, and the agar did not stick to filters. In some of these samples, the agar was released from the filter in the final steps of the staining, resulting in fewer samples for the pre-hurricane data. Due to differences in the level of identification (e.g., taxa vs. morphospecies), for community structure analysis, we used a matrix of taxa classified by phyla. A non-metric multidimensional scaling (nMDS) ordination analysis was used to plot community structure during pre-hurricane, post-hurricane A, and post-hurricane B. For the nMDS, a monthly mean matrix was constructed and built based on a Bray-Curtis matrix. We tested hurricane effects on community structure using a PERMANOVA test and a Bray-Curtis matrix permuted 9999 [57]. A SIMPER test was performed to determine which meiofaunal group were causing the differences in the community structure.

Comparisons of sediment and organic matter percentages between pre-hurricane, post-hurricane A, and post-hurricane B were done with a Kruskal Wallis test due to a lack of normality according to a Shapiro-Wilks test ($p = 0.0001$ for all percentages). Pairwise comparisons of means were made to all sediment and organic matter percentages. A Principal Component Analysis (PCA) [58] was performed to determine which of the environmental variables could explain variations in the meiofaunal community. The PCA was made with ten environmental variables selected based on literature [19,44]. The variables were: Discharge, Nitrate, Phosphate, Silica, Chlorophyll, Coarse Sand, Fine Sand, Organic Matter in Coarse Sand, Organic Matter in Fine Sand, and Percentage of Sediment. We limited the analysis to those ten variables to fulfill the PCA assumption for the appropriate number of observations relative to the analyzed variables [59]. To fulfill this assumption, we did not include the percentage of medium sand, the percentage of organic matter in medium sand, organic matter input, and canopy openness in the PCA analysis. Due to differences in the sampling scales, the PCA matrix consisted of monthly averages of the environmental data. These variables were also standardized to make proper comparisons.

Quantitative models (GLM's) were constructed to determine the relationship between environmental variables and the meiofaunal abundance and richness for post-hurricane A and post-hurricane B. Models for pre-hurricane were published in [46]. Additionally, we made models for the abundance of the two dominant taxa groups (Nematoda and Ciliophora) and dominant feeding groups (bacterivores and predators). Model selection was based on the corrected Akaike's Information Criteria (AICc), which is based on the likelihood of such models corrected for small sample size. We averaged some models due to the possibility of good fits from more than one model (based on delta AICc). Full average models were constructed due to the individual model's low weight [60]. This way, we selected the most parsimonious model that fits our data [61]. All data were revised for the assumptions of the tests.

The nMDS and PCA were run in PAST V 4.05 [62]. PAST was also used for multiple comparisons of the community structure. All other tests were performed using R [63], with the packages Vegan for PERMANOVA tests, MASS for quantitative models, and MuMIn and AICcmodavg for model comparisons and model averages, respectively.

## Results

### Meiofauna

The overall community composition was dominated by nematodes, which composed 35% of the abundance, followed by Ciliophora (18%), Gastrotrich (14%), and Rhizaria (11%). Tardigrada made the smallest contribution to total abundance, accounting for only 2%. The abundance of all taxonomic groups increased during at least one of the post-hurricane periods. Rhizaria and Platyhelminthes were the only two taxonomic groups with post-hurricane B abundances lower than pre-hurricane.

The richness of meiofauna was significantly different among the hurricane periods (H = 35.29, p = 0.001), where post-hurricane A and post-hurricane B had higher richness than pre-hurricane (Fig 1). Richness doubled during post-hurricane A, but decreased by one-third during post-hurricane B. However, all meiofaunal phyla present before the hurricane were present during post-hurricane A and post-hurricane B. Four morphospecies not detected before the hurricanes were observed after the impact of the hurricanes. The new morphospecies belonged to the phylum Ciliophora (1 Litostomatea), Rotifera (*Lecane* like species 1 and 2), and Gastrotricha (*Chaetonotus* species 3, a very large size species).

The confidence intervals (CI, 95%) of meiofauna abundance per $m^2$ showed a large amount of variation in our samples. CI were: 150,280 individuals/$m^2$ (113,491−187,300) for post-hurricane A and 93,094 individuals/$m^2$ (74,312−115,285) for post-hurricane B, while pre hurricane values were between 8,913−59,648 individuals/$m^2$ (3,483−14,405–37,492−92,228). Significant differences in total abundance were detected between pre- and post-hurricane data (H = 52.62, p = 0.001). The highest mean (±SD) abundance was detected during post-hurricane A with 151,058 ± 103,273 individuals/$m^2$ (Fig 1). In general, the most abundant groups immediately after the hurricane were nematodes, followed by testate amoebae, ciliates, and gastrotrichs. Nevertheless, most phyla increased their abundances following the hurricanes (Table 1). The only phyla that did not significantly increase were Amoebozoa, Arthropoda, and Tardigrada (Table 1). In contrast, the abundance of the phyla Rhizaria and Platyhelminthes diminished to pre-hurricane levels during the post-hurricane B period (Table 1).

Meiofauna abundance by feeding groups increased after hurricanes (Table 2). However, only predators continue increasing over time. Herbivores and detritivores increased during post-hurricane A, but their abundances remained similar during post-hurricane B. The abundance of bacterivores and omnivores increased immediately. However, post-hoc pairwise comparisons of abundances between the two post-hurricane periods were not significantly different (Table 2).

The NMDS shows a pattern of post-hurricane changes in community structure (Fig 2). The PERMANOVA test showed that community structure of meiofauna in Quebrada Prieta varied between pre- and post-hurricane periods ($F_{2,31}$ = 8.06, p = 0.001). Trophic structure changed after the hurricanes (Fig 2). The PERMANOVA test showed a significant change between pre- and post-hurricane periods ($F_{2,31}$ = 9.83, p = 0.001). Bacterivores composed 77% of the community, followed by predators with 12%. The least contributing feeding group was detritivores composing 2% of the community. All feeding groups increased their abundance after the hurricane. Predators, herbivores, and detritivores kept increasing in abundance during post-hurricane B. Meanwhile, bacterivores and omnivores reduced their abundances during post-hurricane B, but these abundances were still higher than pre-hurricane.

### Environmental variables

PCA shows a gradient of substrate-related and physicochemical variables (Fig 3). The gradient along axis 1 was associated with the percentages of fine sand, organic matter in fine sand, and coarse sand. Two groups were formed along this axis, one group with a high percentage of fine sand and organic matter in fine sand. A second group is formed with a high

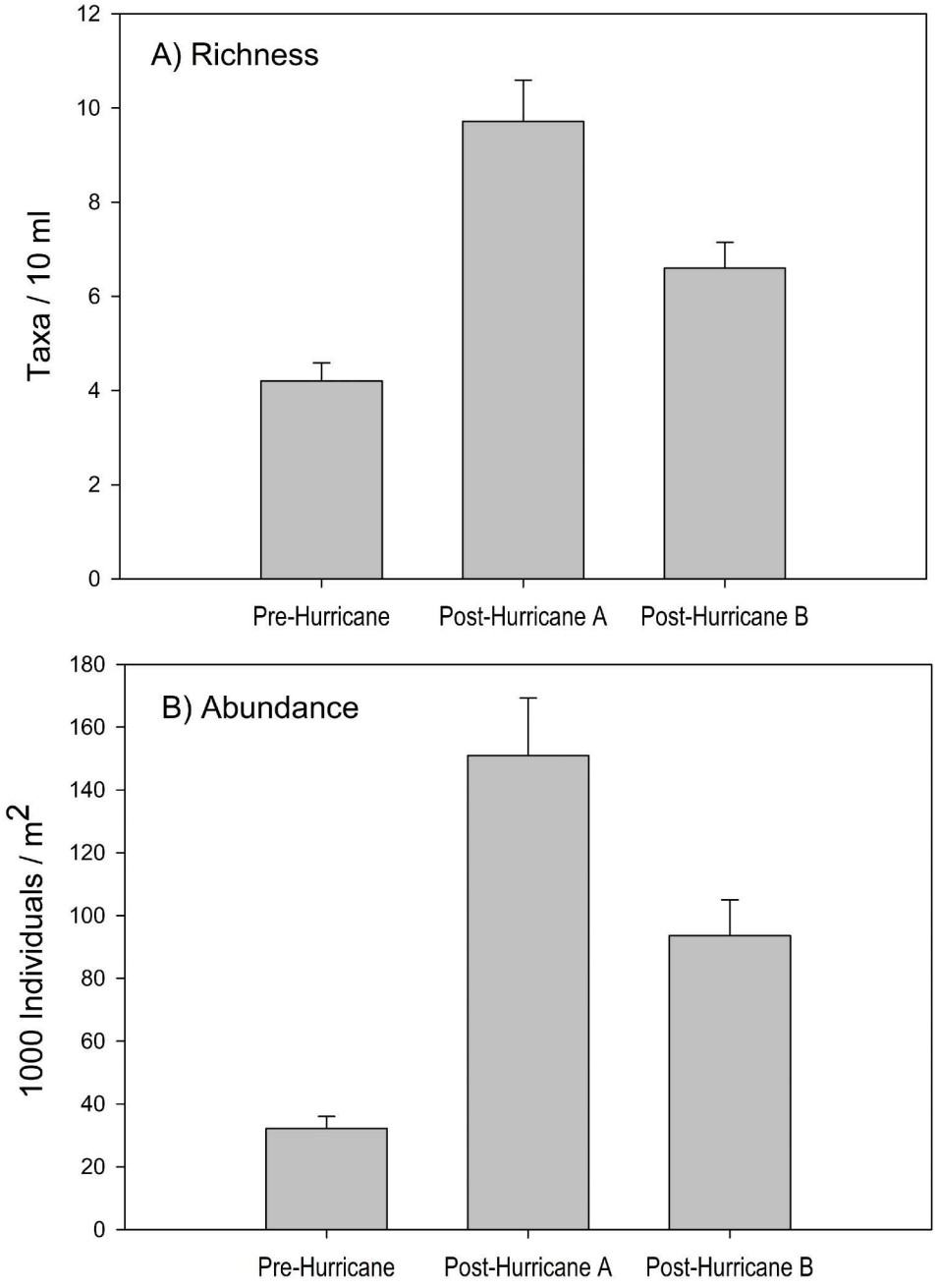

**Fig 1. Changes in richness (A) and abundance (B) of meiofauna during the three study periods in Quebrada Prieta, Luquillo Experimental Forest, Puerto Rico: Pre-Hurricane (January to August 2017), Post-Hurricane A (October to December 2017), and Post-Hurricane B (January to May 2018).** Bars are means±SE of meiofauna richness (taxa/10 mL) and abundance (1000 individuals/ m$^2$).

percentage of coarse sand. For this axis, pre-hurricane data is between post-hurricane A to the right and post-hurricane B to the left. This axis explained 35% of the variability. Along axis 2, environmental data dispersion was caused by other variables: chlorophyll, discharge, and sediment percentage. No groups were formed along this axis; although, this axis explained an additional 16% of the variance.

**Table 1. Mean±SD of meiofauna abundances by taxonomic groups for the three periods studied in Quebrada Prieta, Luquillo Experimental Forest, Puerto Rico. Kruskal Wallis significant p values (<0.05) are highlighted in bold. Means within taxonomic groups with the same letter are not significantly different (p>0.05).**

| Taxonomic group | Hurricane | | | H | p |
|---|---|---|---|---|---|
| | Pre | Post-A | Post-B | | |
| Rhizaria | 9,394 ± 11,575[A] | 24,495 ± 23,639[B] | 7,842 ± 12,289[A] | 17.46 | **0.01** |
| Amoebozoa | 3,188 ± 5,615 | 4,234 ± 7,551 | 3,240 ± 6,013 | 0.06 | 0.96 |
| Ciliophora | 2,732 ± 6,355[A] | 15,423 ± 12,739[B] | 25,612 ± 52,350[B] | 42.73 | **0.01** |
| Arthropoda | 1,878 ± 2,995 | 3,024 ± 6.925 | 2,169 ± 4,071 | 0.13 | 0.90 |
| Nematoda | 6,831 ± 8,620[A] | 62,752 ± 55,558[B] | 30,617 ± 33,699[B] | 42.62 | **0.01** |
| Platyhelminthes | 3,416 ± 6,458[A] | 10,585 ± 11,881[B] | 2,252 ± 3,868[A] | 14.07 | **0.01** |
| Gastrotricha | 2,049 ± 4,860[A] | 16,784 ± 14,799[B] | 14,099 ± 14,538[B] | 43.09 | **0.01** |
| Rotifera | 797 ± 3,241[A] | 8,921 ± 11,986[B] | 2,503 ± 3,976[C] | 22.25 | **0.01** |
| Annelida | 1,708 ± 3,858[A] | 4,990 ± 8,197[B] | 5,5060 ± 10,780[B] | 8.36 | **0.01** |
| Tardigrada | 57 ± 525 | 151 ± 855 | 167 ± 891 | 0.06 | 0.63 |

**Table 2. Mean±SD of meiofauna abundances by feeding groups for the three periods studied in Quebrada Prieta, Luquillo Experimental Forest, Puerto Rico.**

| Feeding group | Hurricane | | | H | p |
|---|---|---|---|---|---|
| | Pre | Post-A | Post-B | | |
| Herbivores | 4,383 ± 6,363[A] | 23,135 ± 16,246[B] | 22,191 ± 27,552[B] | 48.75 | 0.01 |
| Predators | 4,383 ± 6,578[A] | 14,969 ± 14,457[B] | 37,374 ± 43,308[B] | 46.45 | 0.01 |
| Detritivores | 2,163 ± 4,573[A] | 4,890 ± 8,011[B] | 4,088 ± 6,950[B] | 5.25 | 0.02 |
| Omnivores | 1,764 ± 4,147[A] | 8,014 ± 11,955[B] | 2,419 ± 8,563[A] | 10.92 | 0.01 |
| Bacterivores | 18,899 ± 20,001[A] | 100,403 ± 72,378[B] | 28,031 ± 27,383[A] | 40.48 | 0.01 |

Particle size significantly changed after the impact of the hurricanes (Table 3), with post-hurricane samples having a higher medium and fine sand. Coarse sand decreased during post-hurricane A. However, during post-hurricane B, it was similar to pre-hurricane levels (Table 3). Medium sand increased during the post-hurricane B period. Fine sand increased immediately after the hurricane, but during post-hurricane B, it was not different from pre-hurricane values. Organic matter in coarse sand decreased immediately after the hurricane but quickly recovered to pre-hurricane values during post-hurricane B (Table 3). Organic matter in medium sand did not change during post-hurricane A, but increased during post-hurricane B. On the other hand, organic matter in fine sand increased during post-hurricane A, but during post-hurricane B, it went back to pre-hurricane levels (Table 3).

### Relationship between meiofauna and environmental variables

Generalized linear models had the lowest AIC explaining variations in meiofauna abundance (individuals/m²) and richness (taxa/10 mL) in Quebrada Prieta. Models for abundance and richness during post-hurricane A include almost all variables (Table 4). The same result was obtained for major taxonomic groups, where almost all variables were included in the models. In contrast, models for feeding groups were simple (Table 4). Herbivore abundance was positively related to coarse sand and nitrate, and predator abundance was related to organic matter.

For post-hurricane B, models for abundance and richness were simple (Table 4). The abundance of meiofauna was positively related to coarse sand, while a full average model for richness included coarse sand and shrimps as positive

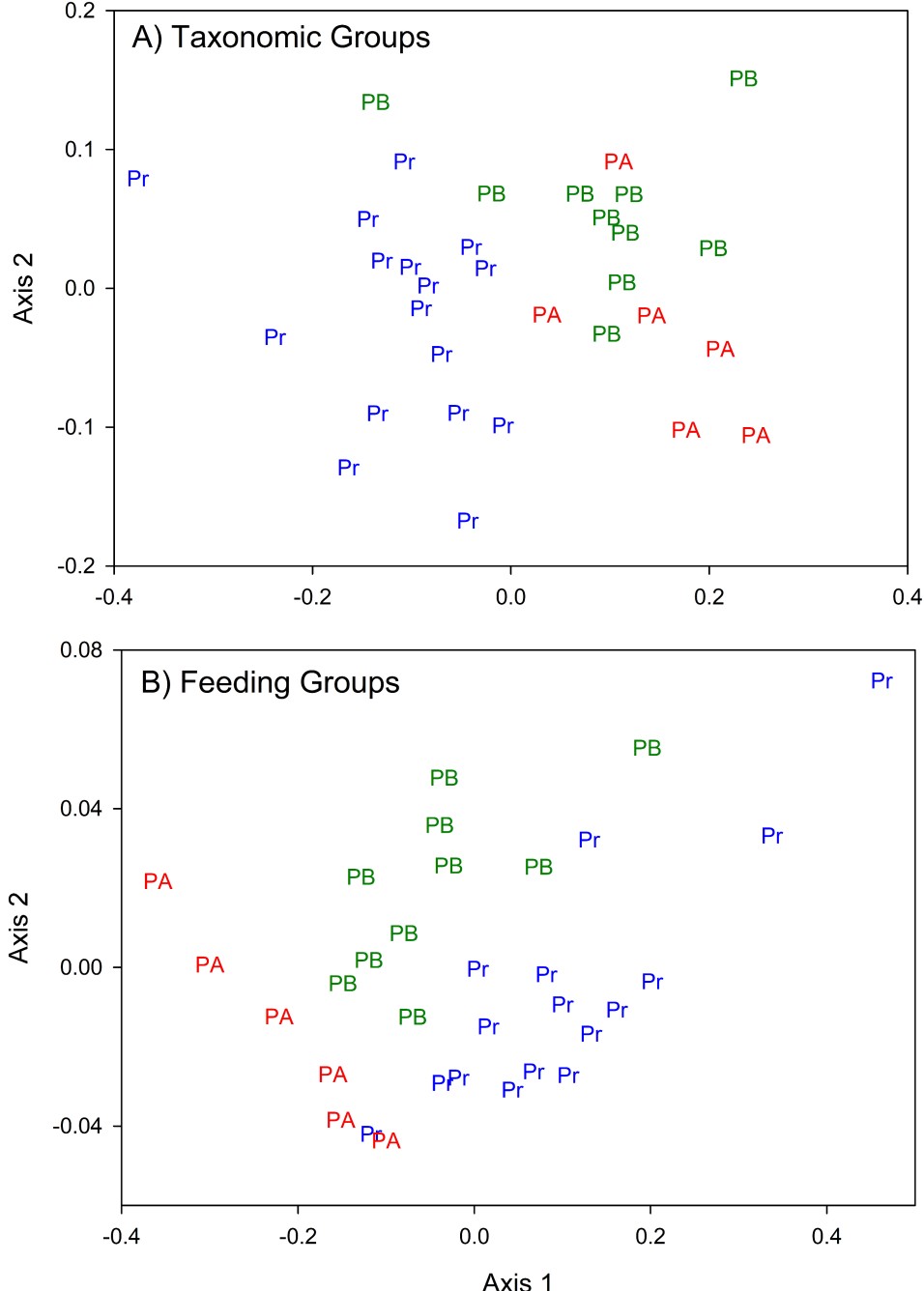

**Fig 2. NMDS showing (A) the meiofauna community taxonomic composition (stress = 0.1) and (B) meiofauna feeding groups (stress = 0.04) in Quebrada Prieta, Luquillo Experimental Forest, Puerto Rico.** The three periods are highlighted: Pre-Hurricane (Pr, blue), Post-Hurricane A (PA, red), and Post-Hurricane B (PB, green).

effects and discharge as a negative effect. The full average model for the abundance of Nematoda was positively related to coarse sand and macroinvertebrates. In comparison, Ciliophora was just related to coarse sand (Table 4). Similarly, herbivores and predators were related to the coarse sand for post-hurricane B.

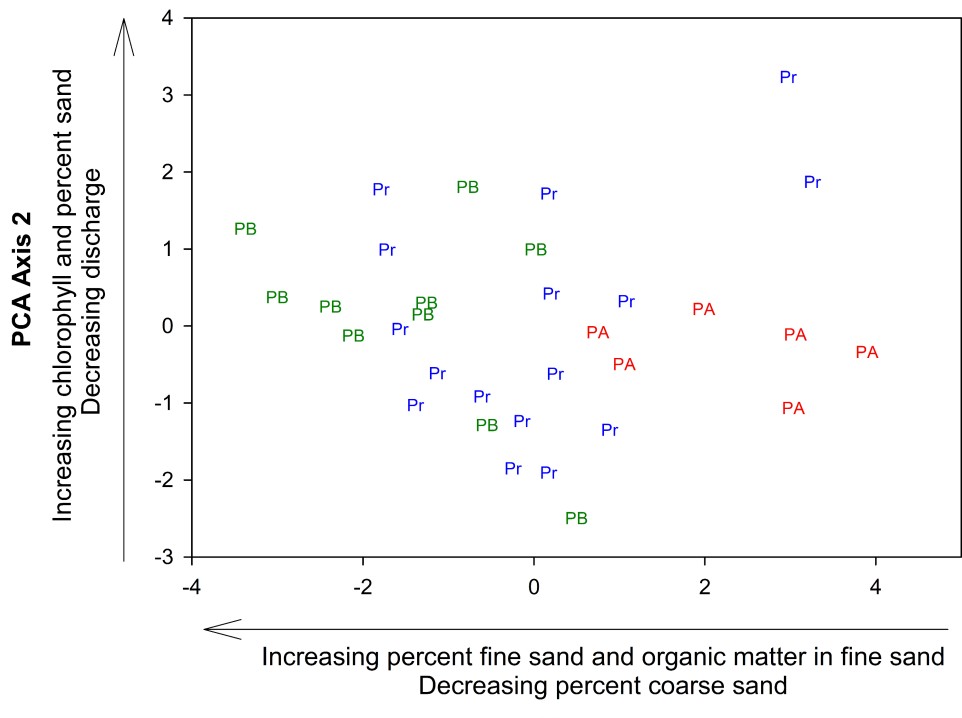

**Fig 3. PCA of environmental variables during pre-hurricane (Pr, blue), post-hurricane A (PA, red), and post-hurricane B (PB, green) showing variations of sediment and discharge-related variables in Quebrada Prieta, Luquillo Experimental Forest, Puerto Rico.** Axis 1 explains 35% of the variation, and Axis 2 explains 16% of the variation.

**Table 3. Mean±SD of environmental variables for the three periods studied in Quebrada Prieta, Luquillo Experimental Forest, Puerto Rico.**

| Environmental variable | Hurricane | | | H | p |
|---|---|---|---|---|---|
| | Pre | Post-A | Post-B | | |
| Coarse Sand | 39±39[B] | 22±22[A] | 43±42[B] | 12.22 | 0.01 |
| Medium Sand | 16±14[A] | 17±15[A] | 23±16[B] | 9.57 | 0.01 |
| Fine Sand | 44±35[A] | 61±30[B] | 35±28[A] | 15.57 | 0.01 |
| Organic Matter in Coarse Sand | 37±29[B] | 21±20[A] | 44±26[B] | 15.48 | 0.01 |
| Organic Matter in Medium Sand | 20±19[A] | 16±13[A] | 23±16[B] | 7.28 | 0.03 |
| Organic Matter in Fine Sand | 42±30[A] | 63±26[B] | 34±24[A] | 23.2 | 0.01 |

## Discussion

### Meiofauna

Similar to the response of most aquatic organisms to disturbance, we expected meiofauna richness and abundance to decrease in response to hurricane impact and then recover as environmental conditions change with time after the disturbance. In contrast, we found that richness and abundance of meiofauna increased. Hurricanes have shaped ecosystems in Puerto Rico for millennia, and the meiofauna of Quebrada Prieta appears to benefit from environmental conditions following this disturbance.

**Table 4. Akaike Information Criterion (AICc) models for richness, abundance, and abundance of Nematoda, Ciliophora, herbivores, and predators during post-hurricane A and post-hurricane B in Quebrada Prieta, Luquillo Experimental Forest, Puerto Rico.**

| Response variable | Model Parameters | AICc | Delta AICc |
|---|---|---|---|
| *Post-Hurricane A* | | | |
| Richness | Shrimp | 34.8 | 0 |
| | Percentage of Coarse Sand | 35.1 | 0.2 |
| | Chlorophyll | 35.4 | 0.6 |
| | Nitrate | 35.4 | 0.6 |
| | Macroinvertebrates | 35.7 | 0.9 |
| | Discharge | 35.9 | 1.0 |
| Model average | Shrimp + Percentage of Coarse Sand + Chlorophyll + Nitrate + Macroinvertebrates + Discharge | | |
| Abundance per m$^2$ | Shrimps + Nitrate + Discharge + Chlorophyll + Macroinvertebrates | 10.0 | 0 |
| | Percentage of Coarse Sand | 158.7 | 148.7 |
| | Nitrate | 163.3 | 153.3 |
| | Discharge | 164.8 | 154.8 |
| | Shrimps | 166.6 | 156.6 |
| | Percentage of Organic Matter | 166.6 | 156.6 |
| Abundance of Nematoda per m$^2$ | Shrimps + Nitrate + Discharge + Chlorophyll + Macroinvertebrates | 4.0 | 0 |
| | Percentage of Coarse Sand | 153.0 | 149.0 |
| | Nitrate | 156.6 | 152.6 |
| | Discharge | 158.4 | 154.4 |
| | Shrimps | 160.6 | 156.6 |
| Abundance of Ciliophora per m$^2$ | Percentage of Coarse Sand | 140.1 | 0 |
| | Percentage of Organic Matter | 140.3 | 0.2 |
| | Shrimps | 140.6 | 0.5 |
| | Nitrate | 140.8 | 0.7 |
| | Chlorophyll | 140.8 | 0.7 |
| | Discharge | 140.8 | 0.7 |
| | Macroinvertebrates | 140.8 | 0.7 |
| Model average | Percentage of Coarse Sand + Percentage of Organic Matter + Shrimps + Nitrate + Chlorophyll + Discharge + Macroinvertebrates | | |
| Abundance of Herbivores per m$^2$ | Percentage of Coarse Sand | 138.1 | 0 |
| | Nitrate | 139.6 | 1.5 |
| | Discharge | 141.0 | 2.9 |
| | Macroinvertebrates | 142.5 | 4.4 |
| | Percentage of Organic Matter | 142.8 | 4.7 |
| Model average | Percentage of Coarse Sand + Nitrate | | |
| Abundance of Predators per m$^2$ | Percentage of Coarse Sand | 135.5 | 0 |
| | Nitrate | 140.6 | 5.1 |
| | Discharge | 140.8 | 5.3 |
| | Shrimp | 142.3 | 6.8 |
| | Chlorophyll | 142.7 | 7.2 |
| *Post-Hurricane B* | | | |
| Richness | Percentage of Coarse Sand | 48.1 | 0 |
| | Shrimps | 48.2 | 0.1 |
| | Discharge | 50.0 | 1.9 |
| | Nitrate | 50.9 | 2.8 |
| | Chlorophyll | 51.6 | 3.6 |
| | Macroinvertebrates | 51.7 | 3.7 |

*(Continued)*

**Table 4.** (Continued)

| Response variable | Model Parameters | AICc | Delta AICc |
|---|---|---|---|
| Model average | Percentage of Coarse Sand + Shrimps + Discharge | | |
| Abundance per m² | Percentage of Coarse Sand | 243.7 | 0 |
| | Shrimps | 284.4 | 4.7 |
| | Discharge + Percentage of Coarse Sand | 248.6 | 4.9 |
| | Discharge | 252.0 | 8.4 |
| | Nitrate | 253.0 | 9.3 |
| | Macroinvertebrates | 253.1 | 9.4 |
| Abundance of Nematoda per m² | Percentage of Coarse Sand | 222.7 | 0 |
| | Macroinvertebrates | 223.5 | 0.8 |
| | Nitrate | 225.2 | 2.5 |
| | Discharge | 225.6 | 2.9 |
| | Chlorophyll | 225.7 | 3.0 |
| Model average | Percentage of Coarse Sand + Macroinvertebrates | | |
| Abundance of Ciliophora per m² | Percentage of Coarse Sand | 222.6 | 0 |
| | Shrimps | 226.1 | 3.5 |
| | Discharge | 228.2 | 5.6 |
| | Percentage of Organic Matter | 230.1 | 7.5 |
| | Percentage of Coarse Sand + Macroinvertebrates + Discharge | 232.9 | 10.3 |
| Abundance of Herbivores per m² | Shrimps | 221.7 | 0 |
| | Percentage of Coarse Sand | 222.1 | 0.5 |
| | Discharge | 224.1 | 2.4 |
| | Percentage of Coarse Sand + Discharge | 224.6 | 2.9 |
| | Percentage of Organic Matter | 226.1 | 4.5 |
| Model average | Shrimps + Percentage of Coarse Sand | | |
| Abundance of Predators per m² | Percentage of Coarse Sand | 227.1 | 0 |
| | Percentage of Coarse Sand + Discharge | 232.1 | 5.0 |
| | Macroinvertebrates | 234.2 | 7.1 |
| | Shrimp | 234.5 | 7.4 |
| | Chlorophyll | 234.8 | 7.7 |

While community richness increased after hurricane disturbance, all major groups were present before and after. The observed increase could be the result of accidental dispersal or the creation of new environmental conditions that permit the establishment of additional species. Increases in richness after hurricanes have been reported. For example, fish richness in the upper zones of the Chesapeake Bay were reported after Hurricane Isabel [64]. However, this increase was due to fish being displaced downstream due to flooding of the Potomac River into the estuarine transition zone. After Hurricane Hugo, a female stink bug *Nezara viridula* (L.) was dispersed over 600 km to Maryland [65]. Furthermore, changes in environmental conditions are common after hurricanes [27,66]. This could explain the increase in species richness of meiofauna. Some meiofaunal groups (e.g., ciliates, testate amoebae, gastrotrichs, tardigrades, and rotifers) can encyst or lay resting eggs [67–70]. These species could be present in the stream, encysted, and thus not detected during surveys. Hurricane disturbance alters the stream environment for several years, potentially causing encysted meiofauna to become active. The new morphospecies detected in Quebrada Prieta after the hurricane were Ciliophora, Rotifera, and Gastrotricha, which are known to encyst [67,68].

New post hurricane conditions have been linked to an increase in the abundance of organisms in our study site [27,66]. Baetid mayflies and shrimp densities increased after hurricane disturbance, potentially due to increased food resources

and debris availability in streams [27,32]. Increases up to 80% have been reported for shrimp densities in streams in LUQ [27]. Similarly, the abundance of coqui frogs after Hurricane Hugo increased due to increased habitat complexity [66]. Meiofaunal increases during the post-hurricane A period could be related to the additional organic matter present, which represented additional resources and habitat. The response of meiofauna in Quebrada Prieta is similar to shrimp responses [27] and coqui frogs [66], and potentially related to increases in food availability (e.g., organic matter, algae) are responsible for the increase in abundance.

The organic matter that enters the stream after the hurricanes is transformed into fine particulate organic matter, which increases available surface area for bacterial colonization [71]. This could explain the increase in bacterivores during post-hurricane [39]. After the hurricanes, this condition changed, and solar radiation reaching the stream increased. This could potentially increase algal productivity. Between 3–7 weeks are required for diatoms to colonize new substrates [72], and post hurricane conditions were favorable for increased algal productivity. This could explain the increase in the abundance of herbivores during the post-hurricane B period. Thus, differences in the taxonomic and feeding community structure between pre-hurricane, post-hurricane A, and post-hurricane B could be associated with food source availability.

Species that benefit from hurricanes are dominant in our study site. Plant communities in LUQ show adaptation to hurricanes, with resistant and resilient species [1,2,12]. In the case of meiofauna, during the post-hurricane B, richness and abundance started to decrease toward pre-hurricane levels, with only the phyla Rhizaria and Platyhelminthes reaching those levels (Table 1). The time to recover to pre-hurricane levels varies by species, ecosystem processes, and ecosystem type [5,7,33,66,73,74]. For instance, nectarivores and frugivorous birds took between 6–10 months for recovery after Hurricane Hugo impacted Puerto Rico. In the case of granivore birds, it took more than one year [74]. In the case of terrestrial invertebrates (e.g., phasmatids, mollusks), recovery took at least 10 months after the impact of the hurricane [75].

## Environmental variables

Changes in meiofauna richness and abundances after the hurricane could be associated with changes in sediment characteristics. Changes in the sediment were expected. Coarse sand during post-hurricane A was very likely washed downstream. Stream discharge during hurricanes will likely have the energy to move large-size particles. However, increases in fine particles could be the result of landslides and tree uproots. This could explain the lack of differences in medium sand during post-hurricane A period when apparently a similar amount of washed-down medium sand was being deposited. Nevertheless, during the post-hurricane B period, deposition of sediment continued, explaining the increase of medium sand during this period.

Tree uprooting is a main event that relocates soils into streams in forested watersheds [76] in combination with landslides, released soils after the hurricanes end in streams, thus increasing sedimentation [9,26,77,78]. This was evident as fine sand increased during the post-hurricane A period. Increments in fine particles were observed after Hurricane Hugo in streams in LUQ [27]. Organic matter in sediments followed a similar pattern. The increase in organic matter in coarse and medium sand could be due to accumulation of organic matter dispersing through the watershed and to fragmentation by macroinvertebrates. Macroinvertebrates are excellent fragmenting organic matter [79,80], especially in headwaters [81].

## Interaction between meiofauna and environmental variables

Stream discharge, coarse sand, and macroinvertebrates were the three main variables associated with changes in meiofauna during the pre-hurricane period in our site [46]. Those variables were also important in shaping communities after the hurricanes. Hurricane floods might not be too different than non-hurricane floods, as evidenced by the impacts of stream discharge on meiofauna (Table 4) [46] and macroinvertebrates [32]. Moreover, it is known that meiofauna possess traits to resist the impact of flooding [20]. However, the relationships between meiofauna and stream discharge, coarse sand, and macroinvertebrates are different between post-hurricane A and post-hurricane B periods. During the post-hurricane period, we also found that nitrate was positively related to meiofauna. The relationship with nitrate could be

more of a correlation than a predictive variable. Nitrate increases after a hurricane are linked to reduced nutrient uptake of plants due to defoliation and uprooting [34].

During the post-hurricane B period, the ecosystem is potentially more stable as it recovered the pre-hurricane levels of rainfall and hydrology. This period is likely to respond to environmental changes in the canopy; thus, models for this period were simple (Table 4). During this period, meiofaunal richness was related to discharge. However, both meiofaunal richness and abundance were related to coarse sand. Similarly, the abundance of meiofauna, ciliates, nematodes, and predators were related to coarse sand. Sediments are a strong variable related to meiofauna elsewhere [18,19,21,46,82]. This relationship was stronger after the hurricane because, during the post-hurricane period B, coarse sand was the only estimator in the models for the abundance of meiofauna, ciliates, and predators.

Macroinvertebrates appear to play an important interaction with meiofaunal communities. Nematodes abundance after the hurricane was related to macroinvertebrate abundance, suggesting a facilitation relationship between both. This could be due to macroinvertebrates fragmenting organic matter. Macroinvertebrates are components of stream ecosystems, where they break down organic matter and facilitate the movement of energy among stream consumers and to the riparian zone [83]. These activities might also benefit meiofaunal communities.

Interestingly, the meiofauna abundance and richness was related to shrimps after the impact of the hurricanes. Although it is known that shrimp can feed on meiofauna [84] and thus regulate their abundance and richness, we did not detect any relationship before the hurricanes [46]. After the hurricanes, shrimp densities are expected to increase [27,33]. This could explain the relationship between shrimps and meiofauna after the hurricanes. Shrimps are likely to influence meiofauna through bioturbation and ingestion of algae and organic matter [85–87].

We did not detect any relationship between meiofauna and chlorophyll during any of the study periods. Massive defoliation of vegetation after hurricanes was a major feature that increased solar radiation in streams. Increases in solar radiation should increase algal growth, a main food resource for meiofauna. The lack of relationship between meiofauna and chlorophyll in Quebrada Prieta could be due to the interaction of stream discharge and shrimps, which prevent the accumulation of algae [88]. Atyid shrimps reduce algal standing crop and determine the composition of algae communities in our study streams, favoring diatoms [86]. Thus, any relationship between chlorophyll and meiofauna in Quebrada Prieta could be overshadow by the grazing activity of shrimps after the hurricane.

## Conclusion

The meiofaunal community in Quebrada Prieta appears to benefit from hurricane disturbances. Hurricanes have affected and shaped ecosystems in Puerto Rico for millennia [1,28]. Quebrada Prieta is located in a steep and rainy area [37,38], and floods are common. Hurricane size floods might not be too different from non-hurricane floods in this area [32]; thus, the meiofauna of Quebrada Prieta could be exploiting these dynamics. Very likely, the meiofauna in Quebrada Prieta has adaptations and traits (e.g., borrowing in sediments, attaching organs, vermiform morphology) to quickly recover from flood disturbance [20]. Similar strategies are used by native stream fauna in the LEF mountains [89].

Something unique to hurricane disturbances is the resulting increase in organic matter in streams from defoliation, debranching, and uprooting of vegetation. The meiofauna in Quebrada Prieta exploits the available resources and overcomes any negative hurricane effect due to increased discharge. This is evident by the increase in abundance, especially bacterivores, during the post-hurricane A period. During this period, organic matter is abundant. As time goes by, the amounts of organic matter are reduced, and then sediments and discharge are again important variables regulating the meiofauna, resulting in the observed abundance during post-hurricane B. We expect that as time goes by, the meiofaunal abundance will continue to diminish until reaching pre-hurricane levels. Long-term studies are needed to confirm this, and Puerto Rico offers an excellent place to conduct such studies [10] on tropical freshwater meiofauna.

## Acknowledgments

We thank the LUQ-LTER for providing support and additional data (StreamFRE members: Bill McDowell, Cathy Pringle, and Pablo Gutiérrez-Fonseca). We also thank Roberto Reyes, Natalia Rodríguez, and the undergrad members of the Ramirez Lab for their help in sampling after the hurricane. Special thanks to Elvira Cuevas for providing access to her laboratory and equipment to process the samples.

## Author contributions

**Conceptualization:** Josué Santiago-Vera, Alonso Ramírez.

**Data curation:** Josué Santiago-Vera, Alonso Ramírez.

**Formal analysis:** Josué Santiago-Vera, Alonso Ramírez.

**Funding acquisition:** Alonso Ramírez.

**Investigation:** Josué Santiago-Vera, Alonso Ramírez.

**Methodology:** Josué Santiago-Vera.

**Writing – original draft:** Josué Santiago-Vera.

**Writing – review & editing:** Josué Santiago-Vera, Alonso Ramírez.

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
