## [Decision Letter · Decision Letter 0]

15 Dec 2025

Dear Dr. Ramírez,

Thank you for submitting your manuscript to PLOS ONE. After careful consideration, we feel that it has merit but does not fully meet PLOS ONE’s publication criteria as it currently stands. Therefore, we invite you to submit a revised version of the manuscript that addresses the points raised during the review process.

Your MS has been reviewed by two reviewers that suggested some minor edits and changes.

We look forward to receiving your revised manuscript.

Kind regards,

Fabrizio Frontalini

Academic Editor

PLOS One

Journal Requirements:

[Support for this work was provided by the National Science Foundation via grant DEB-1831952 (Luquillo Long-Term Ecological Research Program).].

Reviewers' comments:

Reviewer's Responses to Questions

**Comments to the Author**

1. Is the manuscript technically sound, and do the data support the conclusions?

Reviewer #1: Yes

Reviewer #2: Yes

2. Has the statistical analysis been performed appropriately and rigorously?

Reviewer #1: Yes

Reviewer #2: Yes

3. Have the authors made all data underlying the findings in their manuscript fully available?

Reviewer #1: Yes

Reviewer #2: No

4. Is the manuscript presented in an intelligible fashion and written in standard English?

Reviewer #1: Yes

Reviewer #2: Yes

Reviewer #1: This manuscript quantifies the effects of hurricane disturbance on meiofauana composition (abundance and richness) during 8 months pre- and post- Hurricane Irma/Maria. Changes in richness and abundances were linked to organic matter, exhibiting a trajectory towards reaching pre-hurricane levels. The study is important in the context of the existing vegetation bias in hurricane/disturbance studies and in understanding stream dynamics.

The first paragraph begins with hurricane effects on flora, followed by a paragraph describing the effects on fauna. To better establish a link between hurricane effects on plants (that then cascade to faunal responses in the second paragraph), make this connection more explicit, early on in the first paragraph (instead of waiting until L60).

Please define resilient and resistant in L36

L81: Please be more specific about why certain groups would be able to 'take advantage of the new conditions'...is this functional, behavioral, etc.

L81: Similarly, please be more specific about what changes in sediment characteristics have effects on what aspects of change in meiofauna.

L319: It's a stretch to claim that increasing richness and abundances post-hurricane indicate meiofauna are adapted to disturbance. Indeed, the following paragraph outlines other mechanisms (dispersal, environmental niche opening) unrelated to adaptation. Temper this wording (same for the use of adaptation in L353 and L415).

L416: hurricanes have probably shaped ecosystems in Puerto Rico for thousands of years.

Conclusions: New ideas were presented in this paragraph (that hurricane size floods might not be too different from non-hurricane floods; that meiofauna may have adaptations/traits to recovery from hurricane disturbance) but these ideas should also be integrated into the discussion. Then, the conclusion can expand on specific and promising lines of inquiry that future work should focus on.

Figures: The resolution of the figures was low in the reviewer pdf.

Reviewer #2: Abstract –

The use of the term “draining” seems to imply that stream ecosystems are passive and unidirectionally moving downstream. Please replace this term.

Throughout the manuscript there is a mixed use of names to identify the forest where the study takes place. The Luquillo Experimental Forest and El Yunque National Forest occupy the same land footprint, one is not inside the other. Please refer to the Luquillo Experimental Forest to identify your study location, and as a separate note you can state “coterminous” or “co-located” or also known as El Yunque National Forest.

The coarse particulate organic matter (CPOM) noted to increase in the study streams is stated to have come from “hurricane damaged riparian forest”, yet this evidence is not presented. There have been various published studies done in tropical forests that have documented that most CPOM comes flowing from upstream of study reaches.

Introduction –

Please revise the order of the ideas in the Introduction. The paper begins by narrating about hurricanes and trees, not meiofauna. I suggest the authors reconsider in line with the research objectives and order of ideas presented. Share the Why, why is meiofauna important?

Line 36 – Please revise and clarify, is the dense wood only in the stems? Or in the whole tree?

Transitions between ideas within paragraphs should be improved, as in the case of line 39 and 40

Line 43 – The term never should be replaced to “not observed” or “have not recovered” . You can only state about the past and present, you cannot state about the future.

Line 51 – There are more studies of plant communities, more studies of vegetation species. Bias would be to infer and apply what is learned about plant communities to other sets or groups of organisms. Please revise the use of “bias”.

Line 58, 59 and 60 – Please revise the transitions among these ideas and the sentence structure.

Lines 62 to 65 –

State in the initial sentence the reason why Puerto Rico offers an unique opportunity to understand hurricane impacts.

Please revise the idea content and intention to present the ranges of years and scales (Caribbean versus Puerto Rico), how is it that there is no range associated to the “shortened recurrence”, 42 years with no range or error estimation?

Line 66 – The paper cited [39] is from 2004. Is this finding still accurate?

Line 72 and 74 – There are published studies other that [40] and done in tropical forests that have documented multi-year and pre and post hurricane stream CPOM exports and since you state that the pulse of organic matter is important to aquatic fauna, additional evidence from citations would enhance the authors arguments.

Methods

Line 42 – The study cited [41] does not conduct research on the forest cover or tree species, nor does it cite where the statement of “Riparian vegetation is dominated by Dacryodes excelsa (tabonuco) and Prestoea montana (Sierra palm)” comes from. Please cite a primary reference for this statement.

Revise the section from line 98 to 112 – Only hurricanes get names, not tropical depressions. It is not clear why the authors are sharing the wave/depression/storm details before the categorization (and then naming) of the hurricane.

Please revise to add clarity on the combination of the stated “six randomly selected pools” and the “8 samples from each tributary”. Were the “six randomly selected pools” randomized each month that sampling took place? Or just randomly selected initially for the study?

Line 203 – Please clarify what is meant by “monthly differences”.

RESULTS

Line 235 – Very interesting, the implications should be further developed in the Discussion section.

Line 262 – Please remove the word “clear”, that is an interpretation that is not needed.

DISCUSSION

Line 316 – This statement seems to be contradicted with the examples presented in the second paragraph of the Discussion. Please support and revise the ideas.

Line 319- (and elsewhere) replace “centuries” for millennia”.

Line 327 – Use of “on the other hand” seems out place here, as these are not contrasting ideas.

Line 328 – Why “artificially”? Please explain the rationale.

Line 333- There are published studies that document the stream environment altered for several years, not just several months. Please revise.

Please revisit the potential implications stated in the methods where there were fewer samples in the pre-hurricane data.

Line 337- Luquillo Experimental Forest

Line 342 – Concordance, revise to are responsible (replace “is responsible”).

Line 348- explain (remove the s).

Line 352-353 Luquillo Experimental Forest

Line 356 – Varies by individual ecosystem processes, not just by ecosystem and species. This

Line 374 – Luquillo Experimental Forest

Line 401 – What was meiofauna abundance or composition related (related or correlated?) to ? Shrimp abundance? This sentence contains important ideas for the manuscript and needs to be elaborated.

Line 416 – Replace century with millennia

Line 419 – Lugo (2014) has a chapter in a book on disturbance research in Experimental Forests that describes dynamics in the LEF that includes stream response to floods. Perhaps this reference could help elaborate the idea presented here.

Line 422 – Is the implication that there was more hurricane damage along riparian zones than in other parts of the forest? How was it determined that the increase in organic matter that reaches the stream solely from riparian zones?

**Do you want your identity to be public for this peer review?** For information about this choice, including consent withdrawal, please see our Privacy Policy

Reviewer #1: No

Reviewer #2: No

---

## [Author Response · Author response to Decision Letter 1]

2 Feb 2026

See the document attached with detailed answers to each comment made by the reviewers.

---

## [Editor Report · Decision Letter 1]

3 Feb 2026

Hurricane disturbance results in positive effects on tropical stream meiofauna abundance and richness

PONE-D-25-56550R1

Dear Dr. Ramírez,

We’re pleased to inform you that your manuscript has been judged scientifically suitable for publication and will be formally accepted for publication once it meets all outstanding technical requirements.

Kind regards,

Fabrizio Frontalini

Academic Editor

PLOS One
---

## [Editor Report · Acceptance letter]

PONE-D-25-56550R1

PLOS One

Dear Dr. Ramírez,

I'm pleased to inform you that your manuscript has been deemed suitable for publication in PLOS One. Congratulations! Your manuscript is now being handed over to our production team.

Kind regards,

on behalf of

Dr. Fabrizio Frontalini

Academic Editor

PLOS One